# LoCaTE: A Local and Training Dynamics Perspective at Detecting Label Noise in Deep Classification

**A. Anas Chentouf    Haoran Zhang    Marzyeh Ghassemi**
Massachusetts Institute of Technology
{chentouf, haoranz, mghassem}@mit.edu

## Abstract

Noisy labels are a pervasive challenge in modern supervised learning, especially in high-stakes domains like healthcare and moderation, where model reliability is critical. Detecting and mitigating the influence of mislabeled data is essential to improving both performance and interpretability. Building on insights from training dynamics, we propose **Lo**cal **C**onsistency **a**cross **T**raining **E**pochs (LoCaTE), a family of data-filtering methods that leverages over-parameterized neural networks to distinguish clean samples from mislabeled ones. Our approach integrates both local neighborhood information and per-epoch behavior to identify noise and enhance robustness. Evaluated on CIFAR-10/100 under four canonical noise regimes as well as Clothing-1M, LoCaTE achieves competitive $F_1$ scores and improves downstream accuracy by up to seven percentage points. We additionally conduct ablations by studying the performance of LoCaTE on a single epoch. These results highlight LoCaTE as a practical, low-overhead tool for reliable training on noisy datasets.

## 1   Introduction

Supervised learning relies on large, labeled datasets to learn, generalize, and provide useful predictions. The accurate curation of such large datasets is often infeasible. As such, real-life applications use approximate methods to generate labels, ranging from Amazon Mechanical Turk [50] to keyword-based web scraping [3] and the use of pseudo-labels [31]. Unfortunately, prior work has found that such labels are noisy — at least 6% of the labels in ImageNet-1k are incorrect [34]. This is not a unique occurrence: Clothing-1M [50] exhibits an approximated 38% noise rate, while 20% of the labels in WebVision are estimated to be incorrect. The presence of noisy labels in benchmarking datasets not only results in an inaccurate estimation of model performance, but also can destabilize models trained on such data, leading to a significant drop in performance on the true labels [4]. With deep classification models being deployed in safety-critical domains like healthcare, finance, and law [11, 39, 23, 1, 26, 32], it is important to learn models which are robust to noisy labels.

Most work in noisy label detection has followed one of two independent approaches: detecting mislabeled samples and filtering them in downstream training [46, 18, 54], or developing training algorithms which are inherently robust to label noise [12, 24, 35, 28]. In this work, we focus on the former direction, for the following reasons. First, every detection method naturally induces a downstream filtering procedure, allowing us to not only improve final model accuracy but also to rigorously evaluate detection quality. The impact of removal or relabeling can be further studied, e.g. via the use of influence functions [20]. Second, methods to detect and remove mislabeled samples have applications beyond just removing such samples when training downstream models. Such data-filtering methods shed light on tradeoffs between performance and fairness [41], improve future data collection practices, and ensure accurate benchmarking [17]. Finally, by examining which samples are detected as mislabeled, we draw connections to foundational questions in deep learning,

39th Conference on Neural Information Processing Systems (NeurIPS 2025).

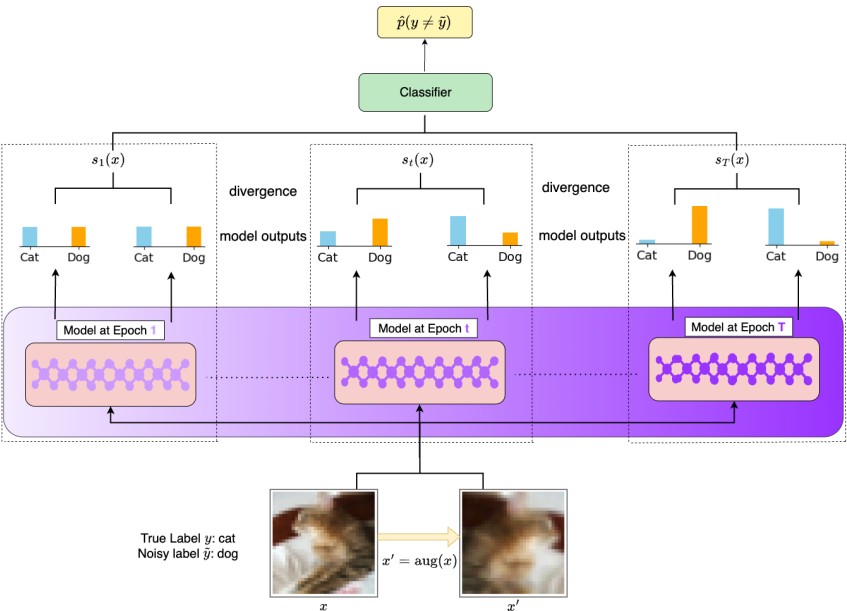

Figure 1: The general workflow for LoCaTE-P. The method first perturbs the sample $x$ using known affine transforms to obtain $x'$ and evaluates the divergence between the model's predictions on $x, x'$ across epochs, then passes this to a simple classifier. A similar procedure is conducted to obtain LoCaTE-M, with a different choice of metric, and where the perturbations $x'$ are the nearest neighbors of $x$.

such as the geometry of the optimization landscape [8, 59] and the generalization behavior of modern neural networks [48].

Existing label noise detection methods can often be computationally expensive, requiring access to pretrained embeddings like CLIP [60] or requiring additional training of complex models. In this work, we provide a lightweight label error detection solution.

Recent filtering work has explored the interaction between memorization and noisy labels [27, 52]. The *memorization* hypothesis posits that over-parametrized neural networks begin by learning the easier, dominant patterns before overfitting to the more difficult patterns. With the intuition that mislabeled samples are more difficult to learn, many methods have been developed to mitigate noisy labels based on training dynamics [56, 55, 27, 25].

We propose a method for noisy label detection based on tracking per-sample neighborhood dynamics during neural network training. **Lo**cal **C**onsistency **a**cross **T**raining **E**pochs (LoCaTE) combines local neighborhood information on training samples across different training epochs. We track two signals: (1) whether a sample's prediction disagrees with the plurality label of its $k$-nearest neighbors in the current logit space, and (2) the degree to which a small augmentation of the input shifts its predicted label distribution. Intuitively, both metrics measure how the model's learned mapping stretches or compresses distances in a sample's immediate neighborhood, either across the nearest neighbors (LoCaTE-M) or under infinitesimal perturbations (LoCaTE-P), thereby quantifying how local distances scale and how decision boundaries evolve in the representation space. As outlined in Figure 1, we record these signals during training. If a small subset ($\leq 5\%$) with clean labels is available, we can fit a lightweight logistic regressor to generate a probability of a sample being mislabeled. We later ablate this clean subset by choosing the epochs of interest heuristically.

We evaluate LoCaTE on four benchmark noise regimes (symmetric, asymmetric, instance, and human) on CIFAR-10/100 [21] using two complementary metrics: $F_1$ for direct label-error detection, and downstream test accuracy when training on a filtered dataset. We then conduct ablations to evaluate the sensitivity of LoCaTE to hyperparameters, including one on Clothing-1M [50] in the absence of clean labels. We find that LoCaTE has competitive $F_1$ scores, while adding comparatively little training overhead and maintaining robustness to hyperparameters.

**Contributions** The contributions of this paper can be summarized as follows.

1. We introduce LoCaTE: a class of easy-to-measure signals integrating local information and training dynamics.

2. We provide both theoretical and conceptual justification for these metrics with connections to the optimization landscape of label noise.

3. We empirically evaluate two instances of LoCaTE on CIFAR-10/100, showing that they establish competitive results in both $F_1$ and downstream accuracy.

4. We demonstrate that LoCaTE is robust to variation in hyperparameters.

## 2 Related Works

The simplest example of label noise is *class-conditioned noise* (CCN), which assumes that the probability that a sample's noisy label depends on its true class and is independent of the instance (the feature): $\Pr[\tilde{Y}|Y, X] = \Pr[\tilde{Y}|Y]$. In other words, the change in labels is governed by a transition matrix $T$ whose entries are $T_{i,j} = \Pr[\tilde{Y} = j|Y = i]$. Other techniques have been proposed to specifically handle the more realistic *instance* noise, where $T = T_x$ varies per instance. We outline some of the aforementioned techniques in the remainder of this section, referring the reader to Song et al. [44], Yuan et al. [55] for a broader overview of literature on noisy labels.

**Loss Functions** In general, one class of methods intended to mitigate label noise design robust loss functions—e.g. symmetric losses are provably tolerant to class-conditioned noise [9] but often exhibit slow convergence. Interpolative losses interpolate between two loss functions: Generalized Cross Entropy (GCE) [58] interpolates between MAE and cross-entropy (CE) via a Box–Cox transform, yielding improved downstream test accuracy. One can also interpolate in a different fashion between MAE and CE, yielding the Generalized Jensen-Shannon loss [7]. Given knowledge of the transition matrix, backward loss correction can "undo" label flips [36], and numerous works aim to estimate $T$ itself [29, 53, 57, 51, 15]. In practice, loss-based methods are easy to implement but suffer from convergence and training difficulties. Moreover, they do not address the direct question of *detecting* and *removing* label noise: is a given sample mislabeled?

**Dynamics-aware Algorithms** Another class of methods that does not detect label noise directly involves dynamics-aware training algorithms: early stopping provably and empirically prevents memorization of noisy labels in over-parameterized nets [25, 56], and tracking how quickly each sample's prediction stabilizes over epochs yields another filter-based approach [55].

**Data Filtering via Training Dynamics** Training dynamics rely on a signal that evolves over training epochs to extract information about training samples. While dynamics-aware training algorithms use signals to inform early stopping, regularization, etc, dynamics can also be used to predict whether a specific sample is mislabeled. Some dynamic-signal methods use training-time margins

$$M_t(x, \tilde{y}) = f_t(x)_{\tilde{y}} - \max_{j \neq \tilde{y}} f_t(x)_j$$

averaged over epochs: AUM thresholds this mean margin to flag noise [37], and DynaCor refines this with discriminative classification [18]. While they are also training dynamics methods, these approaches focus on detection and so they can also be evaluated as binary classifiers (accuracy, AUROC, $F_1$) rather than by downstream test accuracy. Our proposed method LoCaTE falls into this category — leveraging information from across epochs for maximal performance. The multi-epoch condition will be relaxed later, demonstrating competitive performance when only a single epoch is used as well.

**Data Filtering via Clusterability** To detect whether a specific sample is mislabeled, one can look at the sample's nearest neighbors in some appropriate embedding space. Nearest-neighbor-based filters assume clusterability in some embedding: Deep $k$-NN [2] and SimiFeat [61, 60] apply majority voting in logit or pretrained feature space to spot mislabeled samples.

LoCaTE uses a similar framework, aggregating labels from neighbors computed using the model's own logit embeddings. The method's novelty lies in its ability to combine information from different epochs, with the ability to identify critical epochs. By providing a lightweight classifier, not only can LoCaTE be applied on further training instances, but the simple classifier can also be used to gain insights into the interaction between noisy labels and training dynamics. More generally, LoCaTE provides a class of flexible methods that can be fine-tuned for various learning tasks, noise regimes, and datasets.

## 3 Methods

Let $\mathcal{X}$ be our input (feature) space, and let $\mathcal{Y} = \{1, \ldots, C\}$ be our output (label) space. Let $\tilde{D} = \{(x_j, \tilde{y}_j)_{i=1}^{i=n}\}$ be our noisy training set, where $\tilde{y}_j \in \{1, \ldots, C\}$ represents the noisy label. The corresponding true labels are denoted by $y_i$, and also lie in the discrete space $\{1, \ldots, C\}$. This assumption is known as *closed* label noise [45].

The classical supervised learning task is to learn a classifier $f : \mathcal{X} \to \mathcal{Y}$ which minimizes $\mathbb{E}_{(x,y)\sim\mathcal{D}}[\ell(f; x, y)]$. By abuse of notation, our classifier $f$ will produce a probability distribution on $\mathcal{Y}$. In other words, $f : \mathcal{X} \to \Delta^C$. We will use $t$ to denote our training epochs, which range from $t = 1$ to $t = T$. The model $f_t$ is the model obtained after the $t$-th epoch of training. Let $\hat{y}_i^{(t)}$ to denote the prediction of the model $f_t$ of the sample $x_i$:

$$\hat{y}_i^{(t)} = \operatorname{argmax}_j f_t(x)_j.$$

Our proposed signals will take the following general form, capturing information about the training evolution (epochs) per data sample.

**Definition 3.1** (LoCaTE Signal). Let $x$ be a training sample and let $t$ be a training epoch. Given $\mathcal{P}_{t,x}$ a probability distribution on $\mathcal{X}$ and $d$ a metric on $\Delta^C$, we define the training signal associated with $\mathcal{P}_t(x)$ as

$$s_t(x) = \mathbb{E}_{x'\sim\mathcal{P}_t(x)}\big[d\big(f_t(x), f_t(x')\big)\big]. \tag{1}$$

The intention here is that $\mathcal{P}_t(x)$ is a probability distribution that is nearby to $x$ at epoch $t$, which changes dynamically over time. The samples $x' \in \mathcal{P}_t(x)$ are intended to be both semantically and geometrically close to $x$, particularly at certain critical epochs (related to overfitting). For cleanly labeled samples, we expect nearby samples to have nearby predictions. Mislabeled samples often create bubbles of their *noisy* class surrounded by regions of their *true* class [59], and are often found closer to a decision boundary between two classes [42]. As such, we expect the signal values to be significantly higher, provided that the perturbations are not too small.

We note that there is a connection between the perturbations here, and adversarial attacks [10]. However, in our case, these perturbations are intended to be somewhat random. Fawzi et al. [8] show that for random perturbations to elicit similar adversarial phenomena, they would need to be an order of magnitude of the square root of the relevant dimension. Hence, our methodology leverages small, semantic-preserving perturbations — well below the $O(\sqrt{d})$ adversarial threshold, to robustly detect local instability in high-dimensional embedding spaces without inadvertently triggering adversarial effects.

**Definition 3.2** (Dynamic Neighborhood of a Training Sample). Given a training sample $x$ at time $t$, we define its $k$-Neighborhood $\mathcal{N}_{t,k}$ as the $k$ training samples $x' \in \tilde{D}$ with the nearest images to that of $x$, where the images are generated by $f_t$.

Mathematically,

$$\mathcal{N}_{t,k}(x) = \operatorname{argmin}_{x'}^k \|f_t(x) - f_t(x')\|_2, \tag{2}$$

where $f_t$ refers to the model's logits at training epoch $t$.

With this notation, we are ready to introduce our metrics. The first metric, known as the majority metric, focuses on the predictions of the neighbors.

**Definition 3.3** (Majority Metric). We define the majority metric, $\operatorname{maj}_{t,k}(x)$, as the indicator variable of whether the label assigned by $f_t$ to $x$ agrees with the plurality (mode) label among its neighbors. That is,

$$\operatorname{maj}_{t,k}(x) = \mathbf{1}[\tilde{y} \neq \operatorname{mode}\big(\{\hat{y}_{x'}^{(t)} : x' \in \mathcal{N}_{t,k}(x)\}\big)]. \tag{3}$$

Note that this can be clearly written as a LoCaTE signal. Let $\mathcal{P}_{t,k}(x)$ denote the *uniform* distribution over the $k$ nearest neighbors $\mathcal{N}_{t,k}(x)$ of the sample $x$ at epoch $t$. Define the distance metric

$$d_{\mathrm{maj},t,k}\big(f_t(x), f_t(x')\big) \;=\; \mathbf{1}\Big[\arg\max_c f_t^{(c)}(x) \;\neq\; \arg\max_c f_t^{(c)}(x')\Big],$$

where $f_t^{(c)}(x)$ is the logit assigned to class $c$ by the model $f_t$.

With this choice, the generic training–signal template

$$s_t(x) \;=\; \mathbb{E}_{x'\sim\mathcal{P}_{t,k}(x)}\big[d_{\mathrm{maj},t,k}\big(f_t(x), f_t(x')\big)\big]$$

reduces to the *fraction* of neighbors whose predicted label disagrees with that of $x$. We use this continuous version during training as it gives better single-epoch thresholding properties.

**Definition 3.4** (Local Perturbation Metric). Let $G$ be a space of image transformations $g : \mathcal{X} \to \mathcal{X}$. The local perturbations metric is given by

$$\mathrm{per}_{t,G}(x) = \mathbb{E}_{g\in G(x)}\big[\|f_t(x) - f_t(g(x))\|_1\big]. \tag{4}$$

Consider the case where $G$ consists of common image augmentations [16] such as random crops, rotations, and small noises. In this case, $G$ maintains semantic similarity, for many image classification tasks. Hence a perfect model would be invariant under $G$: $f(x) = (f \circ g)(x)$ [22]. One can achieve such invariance by using equivariant networks in the case where $G$ has a group structure [40], or by augmenting the training dataset with elements of its orbit under $G$ [38].

In our case, we apply neither of those strategies, hence the models we train are susceptible to adversarial attacks [10, 30]. The metric $\mathrm{per}_{t,G}(x)$ hence measures the susceptibility of the model to perturbations of the form $G$. In the case where an adversary is allowed to only choose adversarial perturbations generated by $G$, a small value of $\mathrm{per}_{t,G}(x)$ suggests complete invariance. This is used as a proxy to detect label noise.

# 4 Theory

In this section, we provide some theoretical evidence which suggests that the values of our signals are higher for noisy labels. Following Zhu et al. [60], we introduce the assumption of $k$-NN clusterability.

**Assumption 4.1** $((k, \delta_k)$-NN Clusterability). We say that a data set D satisfies the clusterability $(k, \delta_k)$ if for all $x \in D$, the feature $x$ and its $k$ nearest neighbors $x_1, \cdots, x_k$ belong to the same true class with probability at least $1 - \delta_k$.

Because we are interested in the setting of over-trained, over-parameterized networks; we will also assume that the model achieves zero training loss. We discuss how relaxing this assumption affects the results in the appendix.

**Assumption 4.2** (Memorization). For sufficiently large epochs, we assume that $\tilde{y} = \hat{y}$.

**Theorem 4.1.** Assume class-conditioned noise with $T_{i,i} > \frac{1}{2}$. Then, for $\alpha := 1 - T_{i,i}$ denoting the noise rate, we have that

$$\Pr\big[\mathrm{maj}_{t,k}(x) = 1 \,\big|\, \tilde{y} = y\big] \leq \delta_k + \exp\Big(-2k\left(\tfrac{1}{2} - \alpha\right)^2\Big) \tag{5}$$

**Theorem 4.2.** Assume symmetric noise with a noise rate $\alpha < \frac{C-1}{C}$. Then,

$$\Pr\big[\mathrm{maj}_{t,k}(x) = 0 \mid \tilde{y} \neq y\big] \leq \delta_k + \exp\Big(-\tfrac{((C-1)k-\alpha C)^2}{C(C-1)((C-1)k+\alpha C)}\Big) \tag{6}$$

Note that this implies a bound on the AUROC. We defer this result and an extension of theoretical bounds to the perturbation metric, along with proofs, to Appendix A.

# 5 Experiments

Since the intermediate goal is to detect mislabeled samples, we learn a classifier $h$:

$$h : (s_1(x), s_2(x), \cdots, s_T(x)) \mapsto \{0, 1\},$$

where 1 (positive) corresponds to a mislabeled sample and 0 (negative) corresponds to a clean sample.

**Datasets and Noise**   We evaluate our method on CIFAR-10 and CIFAR-100 with four noise types [21], a common setup to evaluate label noise detection and mitigation. The first noise type is symmetric (class-conditioned noise) with $\alpha = 0.6$. The second is asymmetric (class-conditioned noise) with $\alpha = 0.3$, where the transitions are cyclic to the next class: $i \mapsto (i+1) \pmod{C}$. Instance-dependent noise ($\alpha = 0.4$) is generated using a random projection of our image space to capture some features, and human noise ($\alpha \approx 0.09$) is obtained using human annotations [47]. Further details of the noising procedure can be found in the appendix. Later in this section, we also evaluate our methodology on Clothing-1M [50].

**Models and Training**   We train a ResNet-34 [13] on CIFAR-10 and CIFAR-100 for 200 epochs. During training, we record our two signals at every 10 epochs and use them to build a classifier. We also log the signals every epoch for the first 15 epochs; these tend to be extremely informative as we will empirically demonstrate.

For the classifier $h$, we use a simple logistic regression model trained on a small labeled subset of $\tilde{D}$, where we assume access to the true labels $y$ as well. In the upcoming section, we show how we can relax this assumption by choosing a threshold-based classifier at an appropriate epoch.

LoCaTE-M, LoCaTE-P are the method using the majority and perturbation metric, respectively. LoCaTE-M+P is obtained by training a concatenated version of the M, P metrics on a 5% cleanly-labeled validation set. Note that the assumption of having access to a small, cleanly-labeled gold-standard for validation is not uncommon [15, 14], and can be achieved in the active labeling case. An explicit statement of the algorithm is stated in Appendix D.

The perturbation metric LoCaTE-P generates small, semantically-equivalent perturbations of $x$. We do this via image augmentations [16] such as `RandomCrop, RandomFlip, RandomRotate` which are not included as data augmentations when training the model, as well as adding small Gaussian noise with $\mu = 0$ and $\sigma^2 = 0.1$. The can be generalized to general image augmentations.

**Evaluation Metrics**   We evaluate the performance of this classifier in two ways:

1. **Label Error Detection**: We evaluate the $F_1$ score of classification against the true labels.
2. **Downstream Test Accuracy**: We clean the dataset by removing positively-predicted samples and retrain on this new dataset, measuring the downstream test accuracy.

**Baselines**   We compare our method against the following baselines:

- **AUM** [37]: computes the average margin over training epochs, treating persistently low values as a signal of mislabeling. The margin is defined as the logit at the noisy label minus the largest other logit.
- **CL** [33]: estimates a joint distribution between noisy and ground truth labels under CCN assumption, applying the notion of confidence to label quality.
- **Deep $k$-NN** [2]: embeds samples using the model's logits and removes those whose label disagrees with the majority of their $k$ nearest neighbors, thereby mitigating label noise.
- **CORES** [5]: a method of progressively sieving out corrupted examples with a particular choice for training loss.
- **SimiFeat** [60]: extracts pretrained features, then applies $k$-NN majority voting with Bayesian thresholding on those embeddings to score and filter out likely noisy labels. A key factor in this method is the clusterability assumption.
- **DynaCor** [18]: trains an auxiliary classifier on the time-series of per-epoch margins, augmented with synthetic corruptions, to predict whether each sample is clean or mislabeled.

As this is a label noise detection method, we primarily classify using $F_1$ score, defined as the harmonic mean of precision and recall. We favor $F_1$ over raw accuracy because label-noise detection is a highly imbalanced task. A model that trivially predicts "clean" for every sample xcan achieve deceptively high accuracy while failing to retrieve the mislabeled points of interest due to low noise rates; the $F_1$ score penalizes such behavior by weighting precision and recall equally. Accordingly, we report $F_1$ as our primary metric throughout this work, and we also provide the area under the ROC curve (AUROC)

as a complementary, threshold-independent measure of performance, which sheds some light into the distribution of the signals since AUROC = Pr[signal(mislabeled sample) > signal(clean sample)].

## 6 Results

### 6.1 Label Error Detection Performance

In Table 1, we show the results of using LoCaTE to detect label noise in the four benchmark noise regimes. We find that our method is in the top-2 for most noise types, and that it outperforms all baselines in CIFAR-100's instance noise. More generally, we find that our method performs relatively well under instance noise, where mislabeled samples often lie close to decision boundaries between classes [59, 42]. By measuring perturbation divergence—that is, the change in model predictions under semantic-preserving augmentations—we directly quantify the local instability around these boundary points. Likewise, the majority-voting baseline exploits agreement among nearest neighbors in the logit space to capture semantic proximity and flag potential mislabels.

Table 1: $F_1$ score of classification under different noise types on **CIFAR-10** and **CIFAR-100**, reporting mean and the standard deviation computed across 3 random seeds. Results of other methods are obtained from [18, 60]. The baseline row corresponds to the $F_1$ score of the constant classifier, which is equal to $\frac{2\alpha}{1+\alpha}$, where $\alpha$ is the noise rate. The top 2 performing methods (up to significance) for each noise type are **bolded**.

| Method | CIFAR-10 | | | | CIFAR-100 | | | |
|---|---|---|---|---|---|---|---|---|
| | Sym. | Asym. | Inst. | Human | Sym. | Asym. | Inst. | Human |
| Baseline | 75.0 | 56.2 | 57.1 | 16.5 | 75.0 | 56.2 | 57.1 | 16.5 |
| Deep $k$-NN | 82.4 | 75.2 | 63.1 | 56.2 | 70.7 | 56.8 | 63.4 | 57.4 |
| AUM | $75.4 \pm 0.2$ | $46.4 \pm 0.3$ | $57.7 \pm 0.0$ | $16.7 \pm 0.0$ | $75.8 \pm 0.2$ | $46.7 \pm 0.3$ | $57.8 \pm 0.1$ | $58.0 \pm 0.2$ |
| CL | $88.7 \pm 0.6$ | $\mathbf{91.9 \pm 0.1}$ | $82.5 \pm 0.4$ | $57.0 \pm 0.3$ | $77.9 \pm 0.4$ | $62.4 \pm 0.2$ | $67.3 \pm 0.3$ | $65.2 \pm 0.2$ |
| CORES | $\mathbf{92.9 \pm 0.2}$ | $26.7 \pm 0.4$ | $49.2 \pm 1.2$ | $63.6 \pm 0.6$ | $66.3 \pm 0.4$ | $33.8 \pm 0.5$ | $39.2 \pm 0.5$ | $31.9 \pm 0.5$ |
| SimiFeat-V | $\mathbf{94.6 \pm 0.1}$ | $84.7 \pm 0.2$ | $83.7 \pm 0.1$ | $\mathbf{69.4 \pm 0.2}$ | $88.0 \pm 0.1$ | $70.3 \pm 0.1$ | $77.8 \pm 0.1$ | $76.2 \pm 0.1$ |
| SimiFeat-R | $92.9 \pm 1.8$ | $84.0 \pm 0.1$ | $86.9 \pm 0.1$ | $68.8 \pm 0.3$ | $\mathbf{89.7 \pm 0.1}$ | $66.2 \pm 0.1$ | $75.5 \pm 0.1$ | $\mathbf{77.8 \pm 0.1}$ |
| DynaCor | $\mathbf{93.6 \pm 0.2}$ | $\mathbf{94.2 \pm 0.5}$ | $\mathbf{91.5 \pm 0.3}$ | $\mathbf{72.6 \pm 2.5}$ | $\mathbf{91.3 \pm 0.5}$ | $\mathbf{79.2 \pm 0.6}$ | $79.5 \pm 1.1$ | $\mathbf{77.3 \pm 0.5}$ |
| LoCaTE-M | $91.5 \pm 0.3$ | $\mathbf{91.7 \pm 0.4}$ | $90.1 \pm 0.1$ | $64.5 \pm 6.4$ | $89.4 \pm 0.5$ | $\mathbf{83.1 \pm 0.5}$ | $\mathbf{88.7 \pm 0.3}$ | $72.1 \pm 0.3$ |
| LoCaTE-P | $86.9 \pm 0.1$ | $74.0 \pm 2.3$ | $87.7 \pm 0.1$ | $51.6 \pm 0.3$ | $83.4 \pm 0.6$ | $57.9 \pm 0.2$ | $81.6 \pm 0.5$ | $71.5 \pm 0.7$ |
| LoCaTE-M+P | $91.5 \pm 0.0$ | $\mathbf{91.6 \pm 0.2}$ | $90.3 \pm 0.2$ | $62.9 \pm 0.2$ | $89.6 \pm 0.1$ | $\mathbf{82.9 \pm 0.2}$ | $\mathbf{88.5 \pm 0.1}$ | $72.6 \pm 0.2$ |

### 6.2 Accuracy of Downstream Models

One common application of data filtering is to train downstream models on the filtered data. We train models using data filtered from LoCaTE-M on CIFAR-10. In Figure 2, we find that test accuracy increases monotonically until reaching around 60% removal rate, corresponding to the rate of noise in the actual dataset (symmetric noise). This method also achieves competitive downstream training accuracy: removing 60% of samples then training achieves a 7% improvement over using Generalized Cross Entropy's truncated loss trained on the noisy data. A smaller 2% improvement is obtained for pretrained models. Given that the $F_1$ scores are high, removing the highest percentiles of data leads to mostly removing mislabeled samples, and this leads to improved downstream generalization. Past the $\alpha$% point, we begin increasingly removing clean samples, which leads to a drop in performance. When removing almost all data, we see the expected convergence between Cross Entropy and Generalized Cross Entropy.

### 6.3 Single-Epoch LoCaTE

In this section we examine several strategies for selecting a *critical epoch*—a single epoch $t$ at which the signal $s_t(x)$ is measured. As discussed earlier, LoCaTE-M is relatively robust to this choice: with an appropriate epoch, one can attain nearly the same performance (in terms of $F_1$ score) as when aggregating information across all epochs.

Empirically, the epoch that maximizes the $F_1$ score typically coincides with, or lies very close to, the epoch of peak validation accuracy on the (noisy) training data. Figure 4 illustrates this trend by plotting epoch-specific $F_1$ scores over the first 50 training epochs.

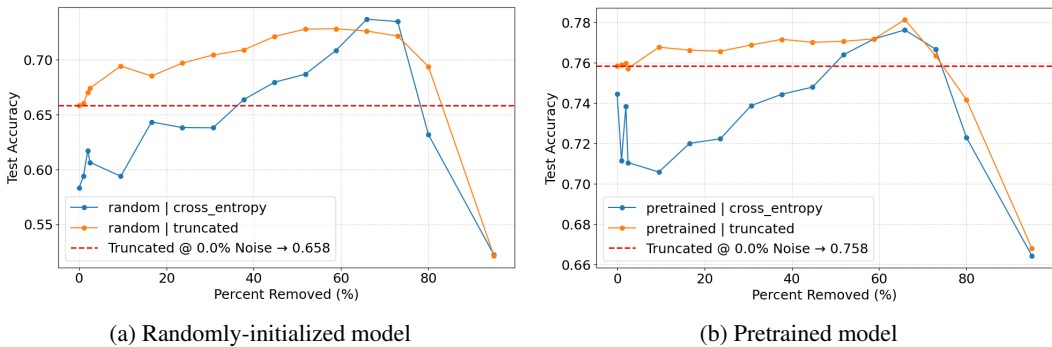

(a) Randomly-initialized model          (b) Pretrained model

Figure 2: Test accuracy after filtering with LoCaTE-M and downstream training on CIFAR-10 with symmetric noise.

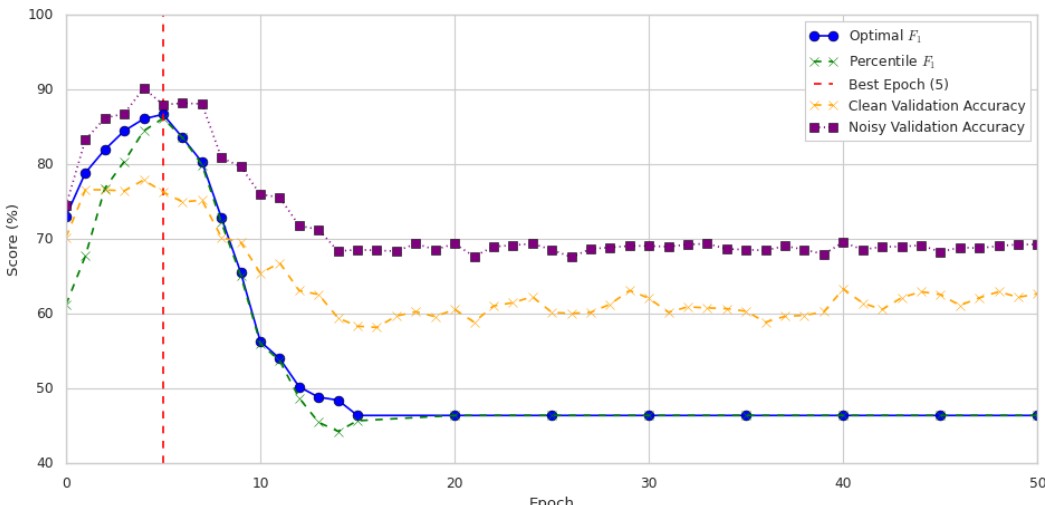

Figure 3: $F_1$ score across epochs for asymmetric noise ($\alpha = 0.3$). The *optimal* $F_1$ is obtained by choosing, for each epoch $t$, the threshold that maximizes the $F_1$ score of $s_t(x)$. The *percentile* $F_1$ is computed by labeling as noisy the top $\alpha\%$ of samples ranked by $s_t(x)$. We restrict $t$ to the range $1 \leq t \leq 50$ and display the validation score across clean and noisy CIFAR-10.

In Table 4 and Table 5, we compare various epoch-selection heuristics for symmetric and asymmetric noise on CIFAR-10 respectively. The *percentile* $F_1$ score is obtained by marking the top $\alpha\%$ of scores as noisy.

Across these noise regimes, selecting the epoch of highest (or second-highest) validation accuracy yields the smallest drop in $F_1$ relative to the full logistic-regression classifier. In other words, a simple validation-based heuristic is sufficient to match LoCaTE-M's performance while avoiding the computational cost of aggregating signals across all epochs. See Appendix D.1 for more details.

### 6.4 No Clean Labels

We also evaluate our method on Clothing-1M [50], a dataset of a million clothing items across 14 classes. The majority of the dataset comes without any clean labels. In fact, the noise rate in Clothing-1M is estimated to be around 38.5% [43]. This experiment allows us to simulate the efficacy of LoCaTE in two important regimes: larger, real-world datasets, as well as the lack of clean labels (we do **not** use the Clothing-1M clean validation subset for training our classifier). We simply average the LoCaTE-M metrics across different epochs, and remove the values with the top $p\%$ values, and then re-train and evaluate the downstream test accuracy on Clothing-1M's clean validation set.

Table 2: CIFAR-10 under asymmetric ($\alpha = 0.3$) and instance ($\alpha = 0.4$) noise. Means $\pm$ s.d.; $\Delta F_1$ is relative to logistic regression on all epochs. The 'Best Epoch' column corresponds to different heuristics for selecting the single best epoch to compute LoCaTE in the absence of labeled data.

| Method | Asymmetric ($\alpha = 0.3$) | | | Instance ($\alpha = 0.4$) | | |
|---|---|---|---|---|---|---|
| | Best Epoch | Percentile $F_1$ (%) | $\Delta F_1$ (%) | Best Epoch | Percentile $F_1$ (%) | $\Delta F_1$ (%) |
| Agreement Max | $40.0 \pm 10.0$ | $46.4 \pm 0.3$ | $-48.1 \pm 0.4$ | $43.3 \pm 7.6$ | $57.1 \pm 0.3$ | $-36.9 \pm 0.4$ |
| Agreement Min | $0.0 \pm 0.0$ | $61.2 \pm 0.7$ | $-31.7 \pm 0.6$ | $0.0 \pm 0.0$ | $85.5 \pm 0.2$ | $-5.5 \pm 0.3$ |
| Entropy Max | $0.0 \pm 0.0$ | $61.2 \pm 0.7$ | $-31.7 \pm 0.6$ | $0.0 \pm 0.0$ | $85.5 \pm 0.2$ | $-5.5 \pm 0.3$ |
| Entropy Min | $40.0 \pm 10.0$ | $46.4 \pm 0.3$ | $-48.1 \pm 0.4$ | $43.3 \pm 7.6$ | $57.1 \pm 0.3$ | $-36.9 \pm 0.4$ |
| First Train Decrease | $17.0 \pm 7.0$ | $46.3 \pm 2.8$ | $-48.3 \pm 3.1$ | $31.7 \pm 7.6$ | $57.1 \pm 0.3$ | $-36.9 \pm 0.4$ |
| Noisy Validation Max | $4.0 \pm 0.0$ | $\mathbf{85.2 \pm 0.6}$ | $-4.9 \pm 0.8$ | $2.7 \pm 0.6$ | $\mathbf{88.8 \pm 0.4}$ | $\mathbf{-1.8 \pm 0.4}$ |
| Noisy Validation Post Max | $5.3 \pm 0.6$ | $\mathbf{84.9 \pm 1.2}$ | $-5.2 \pm 1.5$ | $4.0 \pm 0.0$ | $\mathbf{87.4 \pm 0.5}$ | $-3.3 \pm 0.5$ |
| Clean Validation Max | $2.7 \pm 1.2$ | $79.6 \pm 4.3$ | $-11.1 \pm 4.8$ | $2.7 \pm 0.6$ | $88.8 \pm 0.4$ | $-1.8 \pm 0.4$ |
| Clean Validation Post Max | $4.3 \pm 0.6$ | $85.7 \pm 0.4$ | $-4.2 \pm 0.6$ | $3.7 \pm 0.6$ | $88.0 \pm 0.6$ | $-2.7 \pm 0.6$ |
| Overall Best Percentile $F_1$ | $4.7 \pm 0.6$ | $85.8 \pm 0.4$ | $-4.1 \pm 0.7$ | $2.3 \pm 0.6$ | $88.8 \pm 0.3$ | $-1.8 \pm 0.3$ |
| Logistic Regression | — | $89.5 \pm 0.3$ | — | — | $90.5 \pm 0.1$ | — |

Table 3: Test accuracy of various methods when trained on Clothing-1M using ResNet-50. Results are displayed in mean $\pm$ stdev, with loss correction results taken directly from [36]. LoCaTE-M ($p$) trains a model, removes the samples with the top $p\%$ LoCaTE metrics (averaged acoss epochs), and then retains using CE.

| | Cross Entropy | GCE ($q$=0.7) | Backward $\hat{T}$ | Forward $\hat{T}$ | LoCaTE-M ($p = 10\%$) | LoCaTE-M ($p = 20\%$) | LoCaTE-M ($p = 40\%$) |
|---|---|---|---|---|---|---|---|
| Test Accuracy | $67.9 \pm 0.3$ | $69.0 \pm 0.0$ | $69.1$ | $69.8$ | $\mathbf{69.9} \pm 0.8$ | $69.3 \pm 0.2$ | $69.6 \pm 0.2$ |

While the averaging is a basic priot that does not take into account the results from Section 6.3, the results in Table 3 still demonstrate that LoCaTE outperforms some noise-aware loss functions.

# 7 Conclusion

Our approach connects neighborhood-based voting methods, perturbation-based sensitivity, and training dynamics signals to measure how local distances scale in the learned representation, yielding a robust noisy-label detector. By combining spatial consistency with dynamics over epochs, we achieve $F_1$ performance across four baseline noise patterns, often surpassing much more complex models and methodologies. One advantage of this dual-signal design is that it adapts flexibly to various noise structures and hyperparameters, demonstrating that local curvature in logit space is a powerful indicator of mislabels. Our method is relatively best suited for instance noise, where it relatively performs better than other noise types.

**Limitations** Our pipeline relies on a small clean validation set to train the final logistic-regression classifier, which may introduce labeling costs and risk of misalignment if the validation data poorly reflects the training noise. As a future study, it would be of interest to investigate the impact of using incorrect labels to train this lightweight classifier. In addition to the base model, fitting the LR detector adds a slight computational overhead—both in terms of neighbor searches (and their storage per epoch), but it remains efficient relative to other methods.

**Future Work** Future work could explore relabeling instead of removing, as well as the reweighting data by their likelihood of cleanliness, $1 - h(x)$, instead of completely removing it. A critical question for future work is how noise-mitigation methods affect fairness across subpopulations. Loss-based approaches [58] and early stopping [56] are designed to prevent overfitting to "difficult" or noisy labels, yet those very examples may correspond disproportionately to minority or underrepresented groups. Investigating group-wise performance and developing fairness-aware noise filters will be essential to ensuring equitable model behavior. In the case of LoCaTE-M, we observe that removal of a large percentage of data in CIFAR-10/100 leads to amplifying noise in certain classes. Understanding how noise removal methods create or amplify disparities across classes is an area of future work. Finally, in the absence of clean labels, the methodology in Table 3 averages LoCaTE-M values across epochs. However, as Section 6.3 suggests, there are more optimal weightings that can be deduced without clean labels. We hope to systematically address this question of optimal epoch and percentile selection in future works.

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

## A  Theoretical Results

**Lemma A.1** (Bernoulli's Coupling). Let $q \in (0,1)$. Let $X_1, \cdots, X_k$ are independent Bernoulli variables with parameters $p_i$ such that $q > p_i$ for all $i$, and $S_k = X_1 + \cdots + X_k$ be their sum. Then for all $m$, we have that

$$\Pr[S_k \geq m] \leq \Pr[\text{Bin}(k,q) \geq m].$$

*Proof.* This is a classic application of coupling. Generate $U_1, \cdots, U_n$ i.i.d. drawn from the uniform distribution on $[0,1]$. Set $X_i = \mathbf{1}[U_i \leq p_i]$ and $Y_i = \mathbf{1}[U_i \leq q]$. Note that since $q > p_i$, we have that $X_i \leq Y_i$ a.s. since the event $[U_i \leq p_i]$ is a subset of the event $[U_i \leq q]$. Summing up this inequality, we get that

$$S_k = \sum X_i \leq \sum Y_i \sim \text{Bin}(k,q).$$

Hence for any threshold $m$, the event $[S_k \geq m]$ is a subset of the event $[\text{Bin}(k,q) \geq m]$, and the inequality follows. $\qquad\square$

**Theorem A.2.** Assume class-conditioned noise with $T_{i,i} > \frac{1}{2}$. One has that

$$\Pr\big[\text{maj}_{t,k}(x) = 1 \,\big|\, \tilde{y} = y\big] \leq \delta_k + \exp\left(-2k\left(\tfrac{1}{2} - \alpha\right)^2\right) \tag{7}$$

**Theorem A.3.** Assume symmetric noise with a parameter $\alpha < \frac{C-1}{C}$. One has that

$$\Pr\big[\text{maj}_{t,k}(x) = 0 \,\big|\, \tilde{y} \neq y\big] \leq \delta_k + \exp\left(-\frac{((C-1)k - \alpha C)^2}{C(C-1)\,((C-1)k + \alpha C)}\right) \tag{8}$$

*Proof of Theorem A.2.* Let $X_i = \mathbf{1}[\tilde{y}_i \neq y | \tilde{y} = y]$. Let A be the event that $y = y_1 = \cdots = y_k$. Note that by the union bound, we have

$$\mathbb{E}[X_i|A] = \Pr[\tilde{y}_i \neq y | \tilde{y} = y, A] = \Pr[\tilde{y}_i \neq y_i | \tilde{y} = y, A] \leq \alpha.$$

Moreover, those events are independent, so

$$\Pr[S_k \geq \frac{k}{2} | \tilde{y} = y, A] \leq \exp(-2k(\frac{1}{2} - \alpha)^2).$$

Note also that $\Pr[A^c] \leq \delta_k$ by the clusterability assumption. Combining this with the law of total probability, $\Pr[S_k \geq \frac{k}{2} | \tilde{y} = y]$ can be written as

$$\Pr[S_k \geq \frac{k}{2} | \tilde{y} = y, A]\Pr[A] + \Pr[S_k \geq \frac{k}{2} | \tilde{y} = y, A^c]\Pr[A^c] \leq \exp(-2k(\frac{1}{2} - \alpha)^2) + \delta_k.$$

Finally, observe that $\text{maj}_{t,k}(x) = 1$ implies that $\hat{y}$ is not the mode of its neighbors, and so at least half of the $X_i$'s occured. That is,

$$\Pr[\text{maj}_{t,k}(x) = 1 | \tilde{y} = y] \leq \Pr[S_k \geq \frac{k}{2} | \tilde{y} = y] \leq \exp(-2k(\frac{1}{2} - \alpha)^2) + \delta_k,$$

as desired. $\qquad\square$

*Proof of Theorem A.3.* Similarly, set $X_i = \mathbf{1}[\tilde{y}_i = y | \tilde{y} \neq y]$. Note that $\text{maj}_{t,k}(x) = 0$ implies that $\hat{y}$ is the mode among the neighbors, and so at least $\frac{k}{C}$ of the $X_i$'s must occur. Similarly conditioning on A, we see that the probability of $X_i$'s occurring corresponds to the probability of a label corruption to the class $y$, which, which is upper bounded by $\frac{\alpha}{C-1}$ in the symmetric setting. Since $\frac{\alpha}{C-1} < \frac{1}{C}$, we apply Chernoff to get

$$\Pr\left[ S_k \geq \frac{k}{2} \,\middle|\, \tilde{y} \neq y, A \right] \leq \exp\left( -\frac{((C-1)k - \alpha C)^2}{C(C-1)[(C-1)k + \alpha C]} \right).$$

The $\delta_k$ comes from the law of total expectation, as usual. $\qquad\square$

## A.1 Continuous LoCaTE

We can also relax the assumption and prove an analogous result for another LoCaTE signal. We first define the relaxation of complete memorization.

**Definition A.1** ($\epsilon$-Memorization). We say that model $f$ trained on a dataset $D$ has $\epsilon$-memorized its training dataset $D$ if $\sup_{(x,y) \in D} \|f(x) - y\| \leq \epsilon$.

We consider a continuous version of the previous signals, given by the following definition.

**Definition A.2** (Neighborhood Metric). We define a neighborhood-based distance metric as

$$n_k(x) = \frac{1}{k} \sum_{x_i \in \mathcal{N}_x} \|f(x) - f(x_i)\|. \tag{9}$$

In the following results, we interpret the true and noisy labels $y$ and $\tilde{y}$ as one-hot encodings. The noise in this case in generated by symmetric noise with a parameter $\alpha$.

**Lemma A.4.** Let $D$ be a dataset. Assume that the model $f$ has $\epsilon$-memorized $\tilde{D}$. If $x$ is cleanly labeled (so $\tilde{y} = y$), then for any $s > 0$, we have that

$$\Pr[n_k(x) \geq 2\epsilon + (1+s)\alpha\sqrt{2}] \leq \delta_k + \exp(-2ks^2\alpha^2). \tag{10}$$

*Proof.* By the triangle inequality, we have that

$$n_k(x) = \frac{1}{k} \sum_{x_i \in \mathcal{N}_x} \|f(x) - f(x_i)\| \leq \frac{1}{k} \sum_{x_i \in \mathcal{N}_x} (\|f(x) - \tilde{y}\| + \|\tilde{y} - \tilde{y}_i\| + \|\tilde{y}_i - f(x_i)\|).$$

The first and last terms are upper-bounded by $\epsilon$ each, following the memorization assumption. We further expand the middle term via the triangle inequality:

$$\frac{1}{k} \sum_{x_i \in \mathcal{N}_x} \|\tilde{y} - \tilde{y}_i\| \leq \frac{1}{k} \sum_{x_i \in \mathcal{N}_x} (\|\tilde{y} - y\| + \|y - y_i\| + \|y_i - \tilde{y}_i\|).$$

The first term is zero, since $x$ is a clean label. The second term measures the difference between a point's label and its neighbors' labels, and is controlled by the clusterability assumption. The final term is controlled by the noise rate. This,

$$n_k(x) \leq 2\epsilon + \frac{1}{k} \sum_{x_i \in \mathcal{N}_x} (\|\tilde{y} - y\| + \|y_i - \tilde{y}_i\|).$$

Define $S = \frac{1}{k} \sum_{x_i \in \mathcal{N}_x} (\|\tilde{y} - y\| + \|y_i - \tilde{y}_i\|)$, let $A$ be the event that $y_1, \cdots, y_k$ all equal $y$. Conditioning $S$ on $B$, the second term disappears and we are left with a Bernoulli. Note that $\Pr[A] \leq \delta_k$, and by the law of total expectation:

$$\Pr[n_k(x) \geq 2\epsilon + (1+s)\alpha\sqrt{2}] \leq \delta_k + \Pr[\mathrm{Bin}(k,\alpha) \geq k(1+s)\alpha],$$

which we can upper bound via Chernoff to be

$$\delta_k + \exp(-2ks^2\alpha^2).$$

$\qquad\square$

## A.2 AUROC Bounds

We can also use these results to get AUROC bounds.

**Lemma A.5.** Let $(X, Y)$ be a random pair with $X, Y \in \{0, 1\}$. Define

$$\alpha = \Pr[X = 1 \mid Y = 0], \qquad \beta = \Pr[X = 0 \mid Y = 1].$$

If the classifier's score is the binary variable $X$, then the area under its ROC curve satisfies

$$\text{AUROC} = 1 - \frac{\alpha + \beta}{2}.$$

*Proof.* Take two *independent* copies $(X^{(1)}, Y^{(1)})$ and $(X^{(0)}, Y^{(0)})$ of $(X, Y)$, conditioning on $Y^{(1)} = 1$ and $Y^{(0)} = 0$. By the "probability-of-ranking" definition,

$$\text{AUROC} = \Pr[X^{(1)} > X^{(0)}] + \frac{1}{2} \Pr[X^{(1)} = X^{(0)}].$$

Because $X$ is binary,

$$\Pr[X^{(1)} > X^{(0)}] = (1 - \beta)(1 - \alpha),$$

the tie events are $(1, 1)$ and $(0, 0)$ with probabilities $(1 - \beta)\alpha$ and $\beta(1 - \alpha)$, respectively. Hence

$$\text{AUROC} = (1 - \beta)(1 - \alpha) + \frac{1}{2}\big[(1 - \beta)\alpha + \beta(1 - \alpha)\big] = 1 - \frac{\alpha + \beta}{2}.$$

$\square$

**Corollary A.5.1.** Under symmetric noise with $\alpha < \frac{1}{2}$, one has that

$$\text{AUROC} \geq 1 - \delta_k - \exp\left(-2k(\frac{1}{2} - \alpha)^2\right) - \exp\left(-\frac{\big((C - 1)k - \alpha C\big)^2}{C(C - 1)\big[(C - 1)k + \alpha C\big]}\right).$$

### A.3 Additional Theory

In this subsection, we provide additional theory that relaxes the previous assumptions. Most importantly, we relax the memorization assumption and provide bounds on the performance of LoCaTE relative to the empirical loss of the model.

**Lemma A.6** (CE tail bound). Let $p \in (0, 1]$ be the model probability assigned to the evaluated class and $\ell = -\log p$ its cross-entropy (natural logarithm). If $\mathbb{E}[\ell] \leq \varepsilon$, then for any $\tau \in (0, 1)$,

$$\Pr[p \leq \tau] = \Pr[\ell \geq -\log \tau] \leq \frac{\varepsilon}{-\log \tau}.$$

In particular, for any $\tau \in (1/2, 1)$, $\Pr[p \leq \tau] \leq \varepsilon/(-\log \tau)$ and hence $\Pr[\arg\max_c p_c \neq$ (evaluated class)$] \leq \varepsilon/(-\log \tau)$.

*Proof.* Apply Markov's inequality on the nonnegative $\ell$: for $a > 0$, $\Pr[\ell \geq a] \leq \mathbb{E}[\ell]/a$. Set $a = -\log \tau$. If $p > \tau > 1/2$ then $p > \max_{c \neq y} p_c$ (since $\sum_c p_c = 1$), so the argmax is the evaluated class. $\square$

**Theorem A.7** (Clean false positives under low CE). Assume CCN with per-class noise $\leq \alpha < \frac{1}{2}$ and $(k, \delta_k)$-clusterability. Fix any $\tau \in (1/2, 1)$. Let $L_{\text{clean}} := \mathbb{E}[-\log p_y(X) \mid \text{clean}]$ and let $\bar{L}$ be the empirical training CE at the epoch. Then for any clean sample $(x, y)$,

$$\Pr[\text{maj}_k(x) = 1] \leq \delta_k + \exp\left[-2k\left(\tfrac{1}{2} - q_\tau\right)^2\right], \quad q_\tau := \alpha + \underbrace{(1-\alpha)}_{\text{\% clean labels}} \cdot \frac{L_{\text{clean}}}{-\log \tau} \leq \alpha + \frac{\bar{L}}{-\log \tau}.$$

In particular, if $q_\tau < \frac{1}{2}$, the clean FPR decays as $e^{-\Omega(k)}$.

*Proof.* Let $B$ be the clusterability event that the $k$ neighbors of $x$ all have true label $y$; $\Pr[B] \geq 1 - \delta_k$. Conditional on $B$, a random neighbor $Z$ is clean with probability $\geq 1 - \alpha$ and mislabeled with probability $\leq \alpha$ (by CCN).

For a clean neighbor $Z$, Lemma A.6 with threshold $\tau > 1/2$ gives

$$\Pr[\hat{y}(Z) \neq y \mid \text{clean}] \leq \Pr[p_y(Z) \leq \tau \mid \text{clean}] \leq \frac{L_{\text{clean}}}{-\log \tau}.$$

For a mislabeled neighbor (worst case for us), upper bound the disagreement with $y$ by 1. Therefore, under $B$,

$$\Pr[\hat{y}(Z) \neq y] \leq \alpha \cdot 1 + (1-\alpha) \cdot \frac{L_{\text{clean}}}{-\log \tau} = q_\tau.$$

Assuming conditional independence of the $k$ neighbor predictions given $B$, the number of neighbors disagreeing with $y$ is $\text{Bin}(k, q_\tau)$, hence

$$\Pr[\text{maj}_k(x) = 1 \mid B] = \Pr[\text{Bin}(k, q_\tau) \geq k/2] \leq \exp\left[-2k\left(1/2 - q_\tau\right)^2\right]$$

by Hoeffding. Unconditioning adds the $\delta_k$ term. Finally, $\bar{L} = (1-\alpha)L_{\text{clean}} + \alpha L_{\text{mis}}$ implies $(1-\alpha)L_{\text{clean}} \leq \bar{L}$, yielding the displayed bound. $\square$

## B  Critical Epochs

In this section we examine several strategies for selecting a *critical epoch*—a single epoch $t$ at which the signal $s_t(x)$ is measured. As discussed earlier, LoCaTE-M is highly robust to this choice: with an appropriate epoch, one can attain nearly the same performance (in terms of $F_1$ score) as when aggregating information across all epochs.

Empirically, the epoch that maximizes the $F_1$ score typically coincides with, or lies very close to, the epoch of peak validation accuracy. Figure 4 in the Appendix illustrates this trend by plotting epoch-specific $F_1$ scores over the first 50 training epochs.

We further compare the best single-epoch classifier with the full logistic-regression classifier used in LoCaTE-M. Table 4 and Table 5 report results for symmetric and asymmetric noise, respectively,

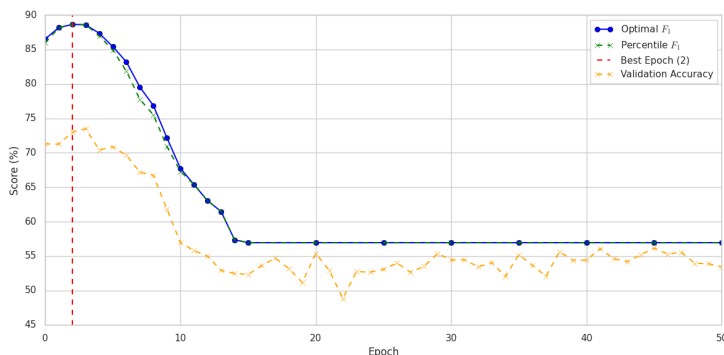

Figure 4: $F_1$ score across epochs. The *optimal* $F_1$ is obtained by choosing, for each epoch $t$, the threshold that maximizes the $F_1$ score of $s_t(x)$. The *percentile* $F_1$ is computed by labeling as noisy the top $\alpha\%$ of samples ranked by $s_t(x)$. We restrict $t$ to the range $1 \leq t \leq 50$.

Table 4: CIFAR-10 (symmetric noise). Means and standard deviations are shown; $\Delta F_1$ is measured relative to the logistic-regression classifier trained on scores from *all* epochs.

| Method | Best Epoch | Percentile $F_1$ (%) | $\Delta F_1$ (%) |
|---|---|---|---|
| Max Agreement of LoCaTE-M | $38.3 \pm 5.8$ | $75.2 \pm 0.3$ | $-17.8 \pm 0.3$ |
| Min Agreement of LoCaTE-M | $0.3 \pm 0.6$ | $86.4 \pm 2.0$ | $-5.6 \pm 2.2$ |
| Max Entropy of LoCaTE-M | $4.3 \pm 2.5$ | $88.0 \pm 2.0$ | $-3.8 \pm 2.1$ |
| First Drop in Train Acc. | $30.3 \pm 22.4$ | $79.5 \pm 7.8$ | $-13.1 \pm 8.5$ |
| Max Validation Acc. | $3.3 \pm 2.3$ | $88.9 \pm 0.5$ | $-2.9 \pm 0.5$ |
| 2nd-Highest Val. Acc. | $5.3 \pm 1.5$ | $89.2 \pm 0.4$ | $-2.5 \pm 0.3$ |
| Logistic Regression | — | $\mathbf{91.5 \pm 0.1}$ | $0$ |

Table 5: CIFAR-10 (asymmetric noise). Means and standard deviations are shown; $\Delta F_1$ is measured relative to the logistic-regression classifier trained on scores from *all* epochs.

| Method | Best Epoch | Percentile $F_1$ (%) | $\Delta F_1$ (%) |
|---|---|---|---|
| Max Agreement of LoCaTE-M | $41.7 \pm 7.6$ | $46.2 \pm 0.2$ | $-48.0 \pm 0.2$ |
| Min Agreement of LoCaTE-M | $0.0 \pm 0.0$ | $59.7 \pm 1.9$ | $-32.9 \pm 2.1$ |
| Max Entropy of LoCaTE-M | $0.0 \pm 0.0$ | $59.7 \pm 1.9$ | $-32.9 \pm 2.1$ |
| Min Entropy of LoCaTE-M | $41.7 \pm 7.6$ | $46.2 \pm 0.2$ | $-48.0 \pm 0.2$ |
| First Drop in Train Acc. | $23.0 \pm 15.7$ | $54.6 \pm 14.7$ | $-38.6 \pm 16.6$ |
| Max Validation Acc. | $2.7 \pm 1.2$ | $79.8 \pm 4.5$ | $-10.3 \pm 4.9$ |
| 2nd-Highest Val. Acc. | $4.3 \pm 1.2$ | $83.1 \pm 2.4$ | $-6.7 \pm 2.6$ |
| Logistic Regression | — | $\mathbf{89.0 \pm 0.1}$ | $0$ |

under a variety of epoch-selection heuristics. The *percentile* $F_1$ score is obtained by marking the top $\alpha\%$ of scores as noisy.

Across both noise regimes, selecting the epoch with the highest (or second-highest) validation accuracy yields the smallest drop in $F_1$ relative to the full logistic-regression classifier. In other words, a simple validation-based heuristic is sufficient to match LoCaTE-M's performance while avoiding the computational cost of aggregating signals across all epochs.

## C Robustness to Hyperparameters

We demonstrate the robustness of our method to the relevant hyperparameters. There are two model-related hyperparameters involved: the number of samples used to train the classifier, and the epochs at which the data is collected.

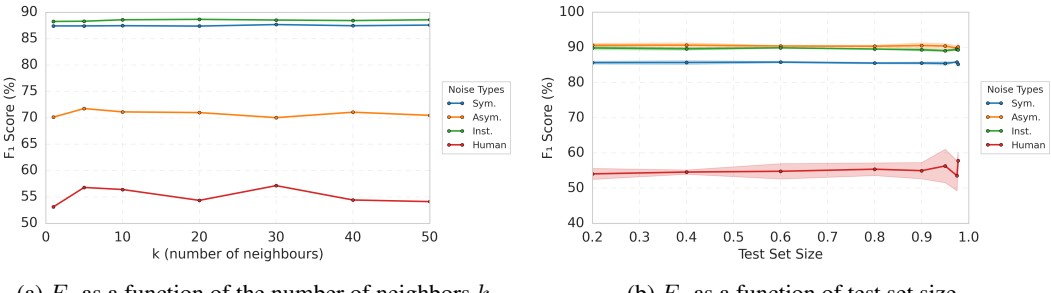

(a) $F_1$ as a function of the number of neighbors $k$.  (b) $F_1$ as a function of test set size.

Figure 5: LoCaTE-M is robust to (a) the choice of $k$ and (b) the amount of gold-standard data available for training the classifier $h$.

Figure 5a shows that increasing the number of nearest neighbors used in LoCaTE-M does not have a significant impact over the $F_1$ obtained: with the largest change observed for human noise with an around 5% increase in $F_1$. Even then, this improvement is not monotonic, suggesting that the hyperparameter $k$ can be fine-tuned to obtain optimal performance. Likewise, Figure 5b demonstrates, with confidence intervals, that training the classifier $h$ on a very small, cleanly-labeled subset of the data, is sufficient.

The use of logistic regression allows user to shed light on the inner-workings of the dataset-specific noise by reviewing the regressor's weights. A natural question to ask is whether a classifier is necessary to achieve this result. In other words, can applying a one-dimensional threshold over our metric at single epoch achieve similar (or better) $F_1$ scores. Our results, summarized in Table 6 for LoCaTE-P, show that an almost identical $F_1$ is obtained at the best epoch. We discuss strategies to discover this best epoch in Section 6.3. This epoch is generally early on, characterizing the critical phase where the model goes from learning new patterns to memorization. Moreover, the primary impact of the classifier is actually on the AUROC, as the $F_1$ often does not improve by much.

Table 6: Epoch-based vs. classifier-based performance of LoCaTE-P across noise types ($F_1$ and AUROC).

| Dataset | Metric | Sym. | Asym. | Inst. | Human |
|---|---|---|---|---|---|
| CIFAR-10 | Best Epoch $F_1$ (%) | $89.5 \pm 0.3$ | $87.7 \pm 0.6$ | $88.5 \pm 0.2$ | $64.4 \pm 5.4$ |
| | Classifier $F_1$ (%) | $91.5 \pm 0.3$ | $91.7 \pm 0.4$ | $90.1 \pm 0.2$ | $64.5 \pm 6.4$ |
| | Absolute $\Delta F_1$ (pp) | $2.0 \pm 0.5$ | $3.9 \pm 0.8$ | $1.7 \pm 0.3$ | $0.1 \pm 1.2$ |
| | Relative $\Delta F_1$ (%) | $2.2 \pm 0.5$ | $4.5 \pm 0.9$ | $1.9 \pm 0.4$ | $0.0 \pm 1.8$ |
| | Best Epoch AUROC (%) | $86.9 \pm 0.2$ | $87.0 \pm 4.4$ | $90.0 \pm 0.1$ | $76.6 \pm 2.7$ |
| | Classifier AUROC (%) | $94.4 \pm 1.2$ | $96.6 \pm 1.7$ | $96.2 \pm 0.8$ | $86.5 \pm 4.5$ |
| | Absolute $\Delta$AUROC (pp) | $7.5 \pm 1.0$ | $9.6 \pm 2.8$ | $6.2 \pm 0.8$ | $9.9 \pm 1.8$ |
| | Relative $\Delta$AUROC (%) | $8.6 \pm 1.1$ | $11.2 \pm 3.7$ | $6.9 \pm 0.9$ | $12.9 \pm 1.8$ |
| CIFAR-100 | Best Epoch $F_1$ (%) | $89.4 \pm 0.3$ | $80.9 \pm 6.1$ | $88.2 \pm 0.1$ | $70.7 \pm 0.1$ |
| | Classifier $F_1$ (%) | $89.9 \pm 1.2$ | $86.0 \pm 5.3$ | $89.2 \pm 1.0$ | $72.1 \pm 0.3$ |
| | Absolute $\Delta F_1$ (pp) | $0.5 \pm 0.9$ | $5.0 \pm 0.9$ | $1.0 \pm 0.9$ | $1.5 \pm 0.2$ |
| | Relative $\Delta F_1$ (%) | $0.6 \pm 1.1$ | $6.3 \pm 1.5$ | $1.2 \pm 1.0$ | $2.1 \pm 0.3$ |
| | Best Epoch AUROC (%) | $85.7 \pm 0.8$ | $91.7 \pm 0.6$ | $90.3 \pm 0.2$ | $79.6 \pm 1.7$ |
| | Classifier AUROC (%) | $95.7 \pm 0.2$ | $98.5 \pm 0.2$ | $97.2 \pm 0.0$ | $91.9 \pm 0.6$ |
| | Absolute $\Delta$AUROC (pp) | $10.0 \pm 0.9$ | $6.8 \pm 0.8$ | $6.9 \pm 0.2$ | $12.3 \pm 2.2$ |
| | Relative $\Delta$AUROC (%) | $11.6 \pm 1.2$ | $7.4 \pm 1.0$ | $7.6 \pm 0.2$ | $15.5 \pm 3.1$ |

These results underscore a practical strength of LoCaTE: its performance is largely insensitive to needing the entire training trajectory. One epoch, chosen carefully, suffices. Because a near-optimal $F_1$ is achieved with a single early checkpoint, practitioners can skip most signal logging, hyperparameter sweeps, and extended over-training, use only a tiny clean subset, and still enjoy similar $F_1$ scores from the logistic regression trained across all epochs. This robustness makes

LoCaTE attractive for real-world pipelines where compute budgets and annotation resources are tight. In the appendix, we detail how this optimal epoch can be found in terms of classic training signals.

# D    Experimental Details

All experiments were conducted using a node of up to 8 A100 GPU's. Unless othewise specified, all standard deviations are computed using 3 random seeds.

## D.1    Algorithmic Description

While we initially described the algorithm, the pseudocode for LoCaTE is provided here.

---

**Algorithm 1** LoCaTE Experimental Pipeline

---

**Require:** Noisy data $\tilde{\mathcal{D}} = \{(x_i, \tilde{y}_i)\}_{i=1}^n$, epochs $T$, log interval $\Delta t = 10$
**Require:** $k$ nearest-neighbors, perturbation family $G$, gold-subset size $p\%$

    **Phase 1 — Backbone training & signal logging**
1: **for** $t \leftarrow 1$ **to** $T$ **do**                                                 ▷ # SGD over noisy data
2:     One epoch of SGD on $\tilde{\mathcal{D}}$                                         ▷ # cross-entropy loss
3:     **if** $t \leq 15$ **or** $t \bmod \Delta t = 0$ **then**                      ▷ # sparse logging schedule
4:         **for all** $(x_i, \tilde{y}_i) \in \tilde{\mathcal{D}}$ **do**                     ▷ # compute per-sample signals
5:             $\hat{y}_i^{(t)} \leftarrow \arg\max_c f_{\theta_t}(x_i)_c$                       ▷ # predicted label
6:             $\mathrm{maj}_{t,k}(x_i) \leftarrow \mathbf{1}\big[\hat{y}_i^{(t)} \neq \mathrm{mode}_{x_j \in \mathcal{N}_{t,k}(x_i)} \hat{y}_j^{(t)}\big]$
7:             $\mathrm{per}_{t,G}(x_i) \leftarrow \mathbb{E}_{g \sim G}\big[\|f_{\theta_t}(x_i) - f_{\theta_t}(gx_i)\|_1\big]$
8:             Store $s_t(x_i) \leftarrow \big(\mathrm{maj}_{t,k}, \mathrm{per}_{t,G}\big)$

    **Phase 2 — Train the noise-detector**
9: **for all** $x_i$ **do**
10:     $\mathbf{s}(x_i) \leftarrow \big[s_{t_1}(x_i), \ldots, s_{t_m}(x_i)\big]$                     ▷ # trajectory features
11: Select $p\%$ gold subset with trusted labels $y_i$
12: Train logistic regression $h$ on $\{(\mathbf{s}(x_i), \mathbf{1}[\tilde{y}_i \neq y_i])\}$           ▷ # predict noise

    **Phase 3 — Clean dataset & retrain**
13: $\mathcal{D}_{\mathrm{clean}} \leftarrow \{(x_i, \tilde{y}_i) \mid h(\mathbf{s}(x_i)) = 0\}$
14: Re-initialize $\theta$ and retrain $f_\theta$ on $\mathcal{D}_{\mathrm{clean}}$
15: $f_{\theta^\star} \leftarrow$ best checkpoint by validation accuracy
        **return** detector $h$, cleaned model $f_{\theta^\star}$

---

## D.2    Hyperparameter Configuration

All runs share a single training recipe so that performance differences are attributable only to the noise settings.

- **Backbone & init.** A `ResNet-34` [13] initialized with ImageNet weights (`initialization=pretrained`).
- **Loss.** Standard cross-entropy (`loss_fn=cross_entropy`).
- **Optimizer.** Adam [19] with learning rate $10^{-3}$ and weight decay 0.001.
- **Training Batch size.** 256.
- **Inference Batch size.** 1024 used for forward passes to compute neighborhoods and LoCaTE signals.
- **Training schedule.** 250 epochs. `epoch_skip=5`, early-epoch cutoff at 15.
- **Embedding.** Dynamic feature obtained by the logit space of the model, with snapshots every `epoch_skip` epochs; computing $k = 50$ neighbors using FAISS [6].

## D.3    Noisy Data

In addition to symmetric and asymmetric noise, which are special classes of class-conditioned noise, we also evaluate our method under two complementary corruption regimes.

**(i) Human noise.** We use the crowdsourced labels released by Wei et al. [47], which capture realistic annotator mistakes.

**(ii) Instance-dependent noise.** Following Xia et al. and Zhu et al. [49, 61], we corrupt an average of $\alpha = 0.4$ of the training labels with the projection-matrix scheme:

1. Draw a per-instance flip probability $q_n \sim \mathcal{N}_{[0,1]}(\eta, 0.1^2)$.
2. Sample a projection matrix $W \in \mathbb{R}^{S \times K}$ with $W_{ij} \sim \mathcal{N}(0, 1)$.
3. Compute class scores $p = q_n \, \mathrm{SoftMax}(x_n W)$, set $p_{y_n} = 0$, and renormalize so $\sum_k p_k = 1$.
4. With probability $1 - q_n$ keep the clean label; otherwise draw the noisy label $\tilde{y}_n \sim \mathrm{Cat}(p)$.

The use of the same projection matrix $W$ implies that similar features have similar noise patterns. For more, see Appendix D from [61].

## D.4  Clothing-1M

For the Clothing-1M experiment, we use the labels from the dataset itself, which are estimated have a noise-rate of 38.5%. This is meant to simulate real-world noise. The experimental procedure is nearly identical, except that we use a `ResNet-34` [13] with a SGD optimizer and a learning rate of $10^{-2}$. Due to the size of the dataset, we only compute the nearest $k = 15$ neighbors.

