# OpenReview forum: "LoCaTE: A Local and Training Dynamics Perspective at Detecting Label Noise in Deep Classification"
_NeurIPS.cc/2025/Workshop/Reliable_ML — NeurIPS 2025 - Reliable ML Workshop_

### Official Review · Reviewer_o7qd · 2025-09-19
**Review for LoCaTE**

**Rating:** 7
**Confidence:** 3

**Review:**

The problem addressed in the paper is important and well-motivated. Noisy labels reduce reliability, and therefore an easy and practical way to spot them is valuable. The main idea of the paper is introducing a new framework: LoCaTE which is clear and intuitive. It suggests looking at local consistency over training epochs using two simple signals: neighbor agreement in logit space (i.e. does a sample’s prediction agree with its neighbors) and sensitivity to small augmentations of the input data. From the simulations provided LoCaTe seems to perform well, especially on instance noise. The method is easy to implement and does not rely on heavy pretrained models. On experiments, in Table 1 the first row “Baseline 75.0/56.2 …”  is unclear; what baseline is that? Also, some rows include mean$\pm$sd while others don’t.  Next, the single epoch selection section is interesting but, several of the other heuristics provided in table 2 perform poorly. It would be helpful to explain why and give further explanations regarding when the critical epoch strategy might fail. Lastly, perhaps it would be useful to try to make reproduction easier by stating k, the distance used on logits etc. Overall, I think this is a solid and practical contribution. The idea is simple, the results are good, and the method is easy to try.

---

### Official Review · Reviewer_sPTq · 2025-09-20
**LoCaTE: A Local and Training Dynamics Perspective at Detecting Label Noise in Deep Classification**

**Rating:** 7
**Confidence:** 2

**Review:**

Summary: The paper tackles noisy-label detection by combining training dynamics with local neighborhood cues. LoCaTE tracks (i) agreement with k-nearest neighbors in logit space and (ii) prediction stability under small augmentations, then fuses these signals (optionally with ~5% clean labels) to flag likely mislabeled samples. It shows strong F1 and better downstream accuracy after filtering across CIFAR-10/100 noise types and a Clothing-1M study, with notable gains on instance noise.

Strengths: This paper addresses a central reliability problem with a simple, low-overhead recipe. Empirical coverage is solid (multiple noise regimes, datasets), and the single-epoch variant is practical while staying close to multi-epoch performance.

Weaknesses: While no clean label is addressed for Clothing-1M, it would be better to add more results of no clean labels as a good part of the results assume reliance on the clean set. Also, it would be great if the author could check on other datasets(vision or non vision datasets).

Suggrstions: Check weaknesses.

Overall, I like the paper and it is completely in scope of the workshop.